# Teleconnection between the climates of the Polar Regions during the last ice age

Xiao Yang<sup>1</sup>, Jose A. Rial<sup>1</sup>

<sup>1</sup>Department of Geological Sciences, University of North Carolina at Chapel Hill, NC-27514, USA

Correspondence to: Xiao Yang (yangxiao@live.unc.edu)

Abstract. Understanding the well-documented differences between Greenland and Antarctica's paleoclimate proxy time series in terms of the dynamic processes connecting the two Polar Regions would help develop a quantitative theory of polar

- climate teleconnections. Multiple conceptual models have been proposed to explain the polar climate time series relationship, and to extend Greenland climate records based on the much longer records from Antarctica. Here we compare the most relevant conceptual models proposed to characterize inter-polar teleconnections associated with these time series. We show that the polar synchronization model, which proposes that the climates of the Polar Regions were phase synchronized over most of the last ice age, shows better overall skill across the range of parameter space under test and
- provides a physical explanation of the polar teleconnection as the mutual synchronization of two nonlinear, coupled oscillators. Phase synchronization results when two or more nonlinear oscillators couple (in this case the two polar climates coupled through the ocean and atmosphere) and therein adjust their (initially different) natural rhythms to a common frequency and constant relative phase. Detailed analyses presented here show that the polar synchronization is a reliable framework to support further studies of polar climate teleconnections.

#### 20 1 Introduction

A number of studies have suggested that the  $\delta^{18}O$  time series of stable isotope temperature proxies from Greenland and Antarctica ice cores are not independent of each other (Blunier et al. 1998; Blunier and Brook 2001; Knutti et al. 2004; Steig 2006; Barker et al. 2009). For instance, the bipolar seesaw hypothesis (Crowley 1992; Broecker 1998; Stocker and Johnsen 2003) states that since abrupt warming episodes in the North Atlantic occur at or near the beginning of gradual cooling in

Antarctica, there must be a strong interaction of the polar climates through meridional heat transport and North Atlantic deep water (NADW) production. EPICA Community Members (2006) argue that for most of the last glaciation there is a 'one-toone' assignment of each Antarctic warming with a corresponding stadial in Greenland. More recently it has been calculated

that the  $\delta^{18}O$  time series from the Polar Regions are phase locked at 90° ( $\pi/2$ ) for most of the last ice age (Figure 1), with Antarctica temperature variations leading Greenland's (Oh et al. 2014).

This phase difference between north and south produces an apparent seesaw, namely, Antarctica warms while Greenland remains cold. From a time series analysis point of view, the polar records form an approximate Hilbert Transform (HT) pair,

- the Greenland record being the HT of the Antarctic record and this in turn the inverse HT of the Greenland's record. Rial (2012) has shown that being a HT pair strongly suggests that the two records are synchronized, thus the polar synchronization hypothesis studied here. Such polar synchronization hypothesis argues that during the last ice age, and likely in earlier times, millennial-scale temperature changes of the north and south Polar Regions were coupled and phase synchronized. Phase synchronization results when two or more nonlinear oscillators couple and therein adjust their (initially
- different) natural rhythms to a common frequency spectrum and constant relative phase (Pikovsky et al. 2001; Balanov et al. 2008).

The foregoing discussion leads to the realization that it is important to determine, with as much clarity as possible, which of the several models proposed to calculate one polar record from the other best satisfies the observations. Knowing which model is better in skills, as well as in which temporal scale they tend to perform better, helps us obtain a more accurate

prediction or hindcast of the polar temperature variation. And such inter-comparison could shed light on the potential mechanism for the interactions of polar climates at various temporal scales. A systematic and quantitative study of the most relevant and popular models was performed and is described presently.

#### 2 Materials and methods

#### 2.1 Models of polar climate teleconnection

Since ice core chronology matching techniques became available, the relationship between polar climates has been of great interest to the paleoclimate community (Bender et al. 1994; Blunier et al. 1998; Blunier and Brook 2001). Although most studies agreed on a one-to-one occurrence of warming events during the last glacial period (Blunier and Brook 2001; Schmittner et al. 2003; EPICA Community Members 2006), the difference in timing and especially in relative phase of these warming events has motivated interest in investigating the mechanism that connects the climates of the Earth's Polar

Regions. Data from ice cores are scarce and limited in resolution, thus simple models have been extensively used to gain information from data and to help identify possible mechanisms. While GCM-based experiments have drawn a close relationship between the bipolar teleconnection and the changes in AMOC (Atlantic Meridional Overturning Circulation) strength (Marcott et al. 2011; Clark et al. 2013), the simplicity of conceptual climate models makes it suitable to test and

5 interpret the essential underlying physics responsible for the teleconnection. On the other end of the spectrum, conceptual models, with few parameters, are ideal in testing many connection mechanisms in a short period of time, guiding the interpretation of climate records and informing the GCM experiments.

Multiple conceptual models have been proposed in the literature based on presumed mechanisms of polar climate teleconnection (Stocker and Johnsen 2003; Schmittner et al. 2003; Rial 2012). However, as will be detailed later, previous

- model studies have used different data pre-treatments, mostly focused on a single direction of record reconstruction. A comprehensive analysis and comparison of the skills of these conceptual models and the models' robustness against parameter change could provide vital information in guiding future climate reconstruction work and improve our understanding of coupling mechanism between polar climates. Furthermore, extending the Greenland climate record beyond the length of its ice core record has been of great interest and all the climate models analyzed here have been used for this
- purpose (Siddall et al. 2006; Barker et al. 2011; Oh et al. 2014). However, this practice was challenged by recent data from high resolution climate records which seem to indicate that the Greenland climate record could be decoupled from the rest of the polar climate system possibly due to the extended sea ice formation during some of the lengthened Greenland stadial (Capron et al. 2010; Landais et al. 2015). It is important to note then that decoupling is not a rare occurrence in synchronized climate oscillators.
- Even though current conceptual models all assume a persistent and constant polar climate teleconnection, a comprehensive inter-comparison study of these models, as presented here, will hopefully help future investigations of the evolution of the polar climate coupling.

# 2.2 Model Intercomparison

The three models studied here can be briefly described as follows: 1) the Thermal Bipolar Seesaw model (TBS) (Stocker and

25 Johnsen 2003), which was built upon the canonical bipolar seesaw concept (Crowley 1992), included the Southern Ocean as

a heat buffer, integrating the heat propagating from the North Atlantic to the Antarctic, until it was recorded in the ice core; 2) An alternative to the TBS was proposed by Rial (2012), who proposes that a phase synchronization (PhaseSync) was at play between polar climate oscillations, with the coupling being the energy transfer through intervening ocean and atmosphere; 3) The Integration/Differentiation model (I/D), the idea of which predates both the TBS and the PhaseSync but

is still in active use (Barker et al. 2011), and uses the simple mathematical transform in its name to relate the polar records. These conceptual models, after being verified by the existing records, have also been used to obtained the first order approximation of Greenland climate history beyond the extend of its ice core record (Siddall et al. 2006; Barker et al. 2011; Oh et al. 2014).

In chronological order, the I/D model was the first conceptual model proposed that accounts for the difference in signal

- shapes between polar records. It was proposed when it became clear that a simple, linear cross-correlation was unable to capture the different signal shapes between polar records (which also means that a simple lead-lag relationship fails to describe the polar climate connection) (see the discussion in (Schmittner et al. 2003; Huybers 2004; Schmittner et al. 2004)). The I/D model states that, with proper trend removal and amplitude adjustments, integrating Greenland's ice core temperature record or differentiating the Antarctic temperature record closely reproduces that of the other pole. This idea
- was revisited in (Rial 2012), in which it was compared to the idea of phase synchronization. And a numeric differentiation was used to reconstruct an 800 kilo-year (ky) millennial scale Greenland climate variation (Barker et al. 2011). Next, the TBS model (Stocker and Johnsen 2003; Stocker 2011) was proposed as an extension to the well-known bipolar seesaw model (Crowley 1992). In the original bipolar seesaw publication, the seesaw behavior was defined as an inverse temperature relationship between the polar regions (Crowley 1992). Conversely, the updated TBS model describes the
- Southern Ocean as a heat reservoir regulating the climate connection between the South Atlantic and the Antarctic. Based on this assumption, Stocker and Johnsen (2003) proposed that the Antarctic temperature variation can be reproduced from the convolution of the Greenland record with an exponential decay term that represents the damping effect of the Southern Ocean. Through trial and error, Stocker and Johnsen determined the characteristic time for the Southern Ocean was ~1120 years, which was consistent in orders of magnitude with that estimated from climate modeling work. Later, Siddall et al.
- (2006)applied the inverse of the TBS model to reconstruct Greenland climate history based on the longer Antarctic record.

More recently, Rial (2012) investigated the phase relationship between the methane-matched polar isotope records and proposed that the polar climate records have been in a state of phase synchronization for most of the last glacial period. While a relatively new idea in interpreting paleoclimate records, phase synchronization is a well-known communication mechanism between pairs of nonlinear oscillators or among a network of oscillators (Pikovsky et al. 2001) [see Figure A1

- for a simple illustration of phase shift]. The phase synchronization as described above is reminiscent of Christiaan Huygens' famous description of the synchronization of two pendulum clocks (analogous to the two polar climates) loosely coupled by minute elastic strain signals (climate signals) sent along the wooden beam (atmosphere and ocean, for instance in the Atlantic basin) on which both hung (Huygens 1986; Pikovsky et al. 2001). In fact, modern models of Huygens' clocks that share the same frequency band show that the phase lock between them can be  $\pi/2$  for a wide range of weak to moderate
- coupling (beam stiffness) (Fradkov and Andrievsky 2007). According to the hypothesis by Rial (2012), the polar records from Greenland and Antarctica are phase locked on the millennial scale with a constant phase difference of  $\pi/2$  for most of the last glacial period, which is a strong evidence of synchronization. This phase difference also explains the difference in signal shapes between the polar records.

The polar climate teleconnection investigated in this study depends heavily on the frequency band chosen (Roe and Steig

- 2004). Before applying the models, the polar records were routinely filtered to isolate the millennial scale frequencies. Furthermore, it is unknown whether or to what degree the changes in data pre-treatment would affect the final skills of the model. As we can see from Table. 1, different studies have different definition of millennial scale variation in terms of what frequency band was to include in the filtered data that would be feed into the models. It is likely that different models will require different frequency bands to achieve optimized skills.
- In addition to differences in data pre-treatment, all these models have been inverted to reproduce the records of another pole (the inverted models are denoted by prefixing "i" to their model names). Individual studies on these models often focused on reproducing the climate record for one of the two poles, with no unified test on the modeling skills when reproducing both poles simultaneously. As invertible climate models that represent the two-way communication of the polar climate system, it is reasonable to expect comparable (or symmetric) skills between reproducing Greenland and Antarctica record. However,
- there is not yet an existing study that conducted this "two-way" test.

In view of above questions, in the following work, we first summarize the filter property of each model by extracting their impulse responses, which, in the case of a linear time-invariant system, should fully describe the system's response to any input. Then we demonstrate and compare the models' ability to reproduce polar climate record based on the record from the opposing pole. Lastly, we test the sensitivity of these models against changing values of data pre-treatment parameters.

Siddall et al. (2006)conducted similar test for the TBS model, where they established that the skill of the TBS depends on both the model parameter  $\tau$  and the filter cutoff frequency  $\sigma$ . To test whether each model gives comparable skills for producing the records of north and south, the sensitivity analysis is carried out in both model directions.

# 2.3 Data

The data used in this study are a  $\delta^{18}O$  proxy from the NGRIP ice core (Greenland) and a  $\delta D$  proxy from the EDC ice core

(Antarctica). Both have been dated with the latest AICC2012 chronology (Veres et al. 2013). Previous modeling practices were based on different north-south matched chronologies, which may affect the comparison between the models. Here, we eliminate such effect from chronology differences by choosing AICC2012 as the proxy chronology, which is considered by most to have the least uncertainty for cross-polar record comparison (Landais et al. 2015).

#### 2.4 Data pre-treatment

- Past polar climate ice core records exhibit a rich spectrum of oscillatory behaviors, from those in the Milankovitch band to those on the millennial, centennial, or weather-caused scale. The cross-pole one-to-one occurrences of the events studied here are of the millennial scale. Thus, before applying the model, the records were normally filtered to isolate the corresponding frequency band. To achieve this, we applied a 4<sup>th</sup> order bandpass Butterworth filter with cutoff frequency from 1/8000 to 1/800 yr<sup>-1</sup>. This effectively suppresses both the direct influence from the long Milankovitch-scale trend and
- the high frequency weather-like signal. The filtered records were then normalized, via division by its own standard deviation after mean removal and tapered to reduce the end effect from the filtering process.

#### 2.5 Transfer functions, convolution, and model characteristics

The I/D, TBS, and PhaseSync models were presented in their time domain forms in their original publications [see Supplemental material for the original model equations], which means that one can implement the model via using polar

climate records as the inputs for their respective formulas. While the time domain model formulas make implementation straight-forward, the filtering effects of each model are more obvious in their frequency domain representations. As the abrupt climate change events have oscillatory behaviors of various scales, it is beneficial to inspect how similar events of different periodicities are affected by each model. One method of doing so is through an inspection of the transfer function

5 of each model. By definition, a transfer function is the response of a system after an impulse signal was given as the input. In the case of the conceptual polar climate models, the transfer function is the model output when a Dirac function was input to the model. The transfer function is an intrinsic property of each model and is independent of the model input (Oppenheim et al. 1996). Once obtained, any model response y(t) can be calculated by convolving the model input x(t) with its transfer function h(t).

$$\mathbf{y}(t) = \mathbf{x}(t) \otimes \mathbf{h}(t) \tag{1}$$

where the  $\otimes$  denotes the convolution operation.

Once calculated, the transfer function of each model was Fourier transformed to obtain the amplitude and phase responses. The transfer functions and their frequency domain representations are summarized in Table 2 and Figure 2 for the Greenland to Antarctica direction. A detailed comparison of the model amplitude and phase responses and its implication will be presented in depth in the following Results section.

### **3 Results**

15

# 3.1 Comparison of amplitude and phase responses

As revealed in Figure 2, both the I/D and TBS models intrinsically function as low-pass filters, which pass the low frequency part of the model input while suppressing the high frequency part. While the two models become identical when  $\tau$  goes to

20 infinity, when the  $\tau$  is in the range of hundreds of years, theoretical cutoff frequencies of the two filters have one order of magnitude difference, such that the I/D model suppresses the high frequency further than the TBS model. However, the actual implementations of the I/D and TBS models yield almost identical results (see simulations in Figure 3a). This similarity is caused by removal of the low frequency portion of the record, where the two filters differ the most, in the data pre-treatment step that was meant to isolate the millennial frequencies. The effect from this initial filter is shown in Figure 2

as the gap at the low frequency region of the amplitude and phase responses, while the green rectangle represents the passing band of each model. The simulation in Figure 3a results from sequentially applying the Butterworth bandpass filter and the polar teleconnection models, showing their combined effect. This combined effect further reveals that the I/D and TBS models share very similar structure in their amplitude responses (see Figure 2 and Figure A2). The results from the

5 PhaseSync model, represented by Hilbert transformation, show that the model functions as an all pass operation that does not change the amplitude of any frequency. However, it does apply a  $\pi/2$  phase shift to the model input, which, in this respect, is the same as the I/D model.

# 3.2 Modeling Antarctic record

With the tapered, normalized, and filtered NGRIP and EDC records, the implementation of each model was carried out by

10 convolving polar record with each transfer function. Then the model outputs are compared to the tapered, normalized, and filtered actual records by calculating the Pearson correlation coefficient between them. The model formulas and their inverted versions can be found in the supplemental information.

We first applied the models to simulate the Antarctic EDC record from that of Greenland's NGRIP (Figure 3a). The crosscorrelation function (CCF) in Figure 3c shows that all three models yielded similar reconstructions, and they all closely

- reproduce the target Antarctic EDC record with maximum correlation values of 0.76, 0.73, 0.73 for the I/D, TBS, and the PhaseSync model respectively. The lags that correspond to maximum correlation in Figure 3c are less than 200 years and are well within the chronology uncertainties of the record. However, the actual EDC  $\delta D$  has distinctive structures that are not replicated by any of the models. For example, the actual record showed complex peak structures (see double/multiple peaks at 54, 39, and 15 kya in EDC in Figure 3a). None of the models investigated here were able to replicate these structures.
- These complex peak structures in the EDC record may result from local climate variations that were independent from the polar climate teleconnection.

#### 3.3 Modeling Greenland record

The model skills differentiate between the PhaseSync and the other two models when reconstructing the Greenland climate from the EDC record (Figure 3b&d). CCF analysis reveals that reconstruction from the PhaseSync model, with the

maximum correlation value of 0.73, more closely simulates the NGRIP record than the reconstructions from either the TBS (0.57) or the I/D model (0.61). The outputs from the I/D and TBS models are similar to each other, but then comparing to the actual NGRIP record, they are not as accurate as the output from the PhaseSync model. The relative amplitudes of the short-duration warming events in MIS3 are over-amplified for the I/D and TBS models when compared to the actual NGRIP

5 record. Also, the lead-lag relations between model results and the target NGRIP record are consistent (Figure 3d), with the model results lagged behind the Greenland record by about 170 years on average (232 years for the TBS model, 150 years for the PhaseSync model, and 134 for the I/D model). Even though this lag may be the consequence of systematic chronology uncertainty, it is consistent with the recent discovery based on WAIS-Divide ice core, which showed abrupt warmings in Greenland on average led those in the Antarctica by 218  $\pm$  92 years (Buizert et al. 2015).

## 10 3.4 Model skill and robustness against data pre-treatment

In order to estimate the skills of these models in reproducing the Antarctic record, we calculated the Pearson correlation coefficients between the simulations and the EDC  $\delta D$  record. As the result of the TBS model depends on the value of the  $\tau$  parameter, we simulated the Antarctic record using different  $\tau$  ranging from 0 to 6000 yrs and plotted the correlation as a function of  $\tau$  (Figure 4a). The correlation coefficients for the other two models are also shown for comparison. The result

15 shows that the skill of the TBS becomes less sensitive to the  $\tau$  value beyond ~1000 yrs, suggesting one might need extra justification for choosing  $\tau$  value in this range. In addition, the tendency of convergence between the TBS and I/D models can be observed from Figure 4a with increasing value of  $\tau$ .

Unlike reproducing the Antarctic record, previous studies have shown that the skill of the TBS model is very sensitive to the change of the smoothing parameter in the data pre-treatment stage, due to the tendency of the TBS and I/D models to

20 amplify high frequency components. Here, this smoothing factor is controlled by the lowpass cutoff frequency of the Butterworth filter (denoted by  $\sigma$ ). To determine the robustness of model skills against changes in this parameter, we have varied the value of  $\sigma$  in the interval between 1/3000 to 1/200 yrs<sup>-1</sup> and plotted the corresponding model skills in Figure 4b for each inverted model. We also fixed the  $\sigma$  value at 1000 yrs and calculated the skill of TBS model with varying values of  $\tau$ . The pair of values that corresponds to the maximum skill of the TBS in Siddall et al. (2006) was  $\sigma = 1000$  yrs and  $\tau =$ 

## 25 500 yrs. However, since we are using a different data set with a different chronology [see (Stocker and Johnsen 2003) for

data used there], we do not expect to obtain the exact same optimal values. We are aware of the potential differences in modeling results that may be due to the differences in the chronologies used. Since we are using the same record pair and the same chronology consistently throughout this study, such effect is minimized.

The results shown in Figure 4b-c suggest that in order to choose the corner frequency of the low-pass filter  $\sigma$ , one has to

- compromise between model resolution and the noise-to-signal ratio. The higher the  $\sigma$  value, the higher frequency variations will be included in the input signal. This means that the models are more likely to be able to replicate the fine details of the abrupt northern climate variations. However, using a high value of  $\sigma$  leads to the model results being dominated by noise. This is especially the case for I/D and TBS models. In the extreme case of no low-pass filter used, the results of the TBS and I/D models are close to pure noise (see Figure A3 in the Appendix). We used  $\sigma = 1/800$  yr<sup>-1</sup> for the simulations in Figure
- 3b (a similar value to 1/750 yr<sup>-1</sup> used in (Barker et al. 2011)). It is worth noting that a proper low-pass filter will also increase the skill of the PhaseSync model. But such a filter is not necessary for the PhaseSync model to perform well (the correlation coefficient between the NGRIP record and the simulation from the PhaseSync model with no low-pass filter is about 0.5).

So far, we have summarized the robustness of the model skills in reproducing the original record of a single pole. To

- conclude, with the same dataset and data pre-treatment, we have compiled the model skills in reproducing both polar records against changing value of  $\sigma$  (see Figure 5). As shown in the figure, the PhaseSync is the most robust of the three models, as its skill in reproducing both polar records stays the highest for almost all values of  $\sigma$ . Yet it yields comparable skills for reproducing the records from both poles (notice that the results from PhaseSync are close to the line of slope = 1). Conversely, the I/D and TBS models are slightly better at reproducing the Antarctic record than the PhaseSync model for
- some low values of  $\sigma$ , but their skills differ greatly in their ability to reproduce the Greenland records and their ability to reproduce the Antarctic records. Specifically, as can been seen from Figure 5, their skills in the Antarctica-to-Greenland direction are noticeably sensitive to changes in parameter  $\sigma$  in the range of 1/3000 to 1/200 yrs<sup>-1</sup>.

#### 4 Discussion

Rial (2012) showed that the HT of the Antarctic record closely reproduces the Greenland record, and conversely, the inverse

Hilbert transform of the Greenland record reproduces the Antarctic record. Here we have also shown that the HT polar synchronization model performs much better than any of its main competitors, I/D and TBS models.

Polar synchronization explains why Heinrich events (Bond and Lotti 1995; Hemming 2004) are likely produced by bipolar dynamics. In fact, Yang et al. (2014) have shown the polar  $\pi/2$  phase shift is climatically important because it makes the

- coldest times of the Greenland temperature nearly coincide with the peak warming events of Antarctica (within age uncertainty), which causes the S-N temperature difference between poles to reach maxima of +10°C to +15°C during these times. More importantly, they show that these maxima are coeval (within age uncertainty) with the Heinrich events (HEs) as described and timed by Hemming (2004) and the quasi-periodic ~1.5ky smaller pulses of IRDs described and timed by Bond and Lotti (1995).
- The I/D and TBS models share many numeric properties, which can be explained by the fact that the latter converges to the former when its  $\tau$  parameter approaches infinity. Both behave as low-pass/high-pass filters in the Greenland-to-Antarctica/Antarctica-to-Greenland directions, respectively. Furthermore, their skills are both asymmetric and sensitive to the inclusion of high frequency content in the model input (Figure 5). But, while the TBS and I/D models selectively suppress the high frequency content of the input when reproducing Antarctic climate (or the low frequency with their
- inverse), the PhaseSync model passes all the frequency content. Because of this property, the PhaseSync model is able to reproduce high frequency climate variations when used to model the Greenland record. And in contrast to I/D and TBS models, its skill is directionally symmetric (that is, comparable skills under the same data pre-treatment) in reproducing polar records.

Both the TBS and PhaseSync models were originally built upon assumptions of specific physical processes. The TBS model

- attributed the polar climates coupling to the heat transfer in the Atlantic Ocean. In its original form, Stocker and Johnsen (2003) derived the Antarctic temperature evolution as the result of heat transfer from the north, thus a north-to-south one way coupling. In contrast, the phase synchronization model requires a bi-directional coupling between the polar climate oscillations. Although it does not inherently restrict the exact form of coupling, the intervening oceanic and atmospheric processes provide various potential mechanisms of coupling. A particular form of the polar climate coupling proposed by
- Rial (2012) was based on two coupled nonlinear Van der Pol oscillators. The van der Pol-Duffing oscillator was originally

proposed by Saltzman (1982; 2002) as the simplest model of polar interaction between sea ice extent and mean ocean temperature. In this specific model setting, the polar climate oscillators, being coupled through the difference in the mean temperature of the ocean between the northern and southern hemispheres, is able to reproduce both the characteristics of polar records and their coupling. This provides a relationship between the poles that involves the mutual interaction between

5 the two Polar Regions, an appealing mechanism based in well-known properties of nonlinear coupled synchronizing oscillators.

In conclusion, polar teleconnection conceptual models provide a mathematical description of the relationship between ice core records, at the same time, they also suggest potential physical coupling mechanisms. Our analysis of these models find that the PhaseSync model outperforms the I/D and TBS models for a wide range of scale of temporal variability in the polar

- records, closely reproducing climate records from both poles. The inter-comparison among models introduced the merits of phase synchronization as a framework for the polar climate teleconnection and suggest that the oceanic heat transfer as described in the thermal bipolar seesaw conceptual model could be integrated into the synchronization framework as a mechanism of coupling between the polar climate oscillations. Future studies will try to establish the evolution of the coupling strength between the polar climates and utilize that information to assess reconstruction and extension of polar
- climate based on the conceptual models.

#### Acknowledgements

The ice core data used in this study can be obtained through the National Oceanic and Atmospheric Administration (NOAA) online database. This research is supported by grants from the J.S. McDonnell Foundation (21st Century Science Initiative on Complex Systems), the National Science Foundation (Paleoclimate and P2C2 programs), and the Martin Fund from the

20 Department of Geological Sciences, University of North Carolina at Chapel Hill. We have no conflicts of interest to disclose.

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

# Appendices

# 10 A. Model details

# Integration/Differentiation model (I/D)

The conceptual model used in (Schmittner et al. 2003) is the following

$$\frac{\partial A(t)}{\partial t} = -sG(t)$$

converting it to discrete form

15

$$A(i) = \sum_{k=0}^{i} G(k) * dt$$

$$G(i) = (A(i) - A(i-1))/dt$$

#### Thermal Bipolar Seesaw model (TBS)

The model presented in (Stocker and Johnsen 2003)

$$A(t) = -\frac{1}{\tau} \int_0^t G(t-t') \exp(-t'/\tau) dt' + A(0) \exp(-t/\tau)$$

20 converting to its discrete form

$$A(i) = -\frac{1}{\tau} \sum_{k=0}^{i} G(i-k) \exp(-k/\tau) + A(0) \exp(-i/\tau)$$

$$G(i) = -A(i) - \tau \frac{A(i-1) - A(i)}{dt}$$

## Phase Synchronization model (PhaseSync)

$$A(i) = G(i) \otimes h_{G \to A}(i)$$
$$G(i) = A(i) \otimes h_{A \to G}(i)$$

For even number of points:

$$h_{G\to A}(i) = \frac{2}{N}\sin(\frac{i}{2}\pi)^2\cot(\frac{i}{N}\pi)$$

For odd number of points:

$$h_{G \to A}(i) = \frac{1}{N} \left[ \cot(\frac{i}{N}\pi) - \frac{\cos(\pi)}{\sin(\frac{i}{N}\pi)} \right]$$

In these equations,

- A(t): the Antarctica climate signal. A(i) is its discrete form.
  - G(t): the Greenland climate signal. G(i) is its discrete form.
  - -G(i): the climate signal in the South Atlantic according to bipolar seesaw model.
  - *τ*: the characteristic timescale.
  - $\otimes$ : convolution symbol.

# B.

This section of appendix contains three supporting figures. Figure A1 is a demonstration of the concept of phase shift, as compared to the more widely known time shift. Figure A2 is an extension of the Figure 2 in the main text. It plots the amplitude and phase responses of the I/D and TBS models together to aid comparison and to stress their similarity. Figure

A3, similar to Figure 3b in the main text, is a reconstruction of the Greenland climate without using any lowpass filters before applying models. The results in Figure A3 demonstrate the necessity of removing high frequencies for the I/D and TBS model to yield meaningful reconstructions. In contrast, the PhaseSync model was less affected by the exclusion of the low-pass filter.

Figure 1: Constant, π/2 phase difference between north and south records during most of the ice age are an indication of polar synchronization. Full-circle 2π phase slippages are likely caused by noise or temporary de-synchronization. Phase difference of π/2
is clearer with the 2π slippages removed, as shown in the bottom plots in (A) and (B). Chronologies are based on methane-matched Monte Carlo age models of δ<sup>18</sup>O proxies NGRIP (Greenland) and Dome C (Antarctica) in (A) and GRIP (Greenland) and Byrd (Antarctica) in (B). Phase differences are calculated using the Empirical Mode Decomposition (EMD) described in (Huang et al. 1998) which allows decomposition of each series into Intrinsic Mode Functions (IMFs). Gabor's method (Gabor 1946; Rial 2012; Rial et al. 2013) has been used to obtain the phase. Note that on the short term the phase difference may be different from π/2 but the long-term average is predominantly π/2 as shown in the histograms. Modified from Oh et al. (2014).

Figure 2 Model characteristics. *Left column*: time domain representations of the model transfer functions.  $\tau = 1120$  years is used to calculate the transfer function for TBS model. *Middle column*: amplitude responses of the transfer functions. The green portion spans the passband of the amplitude responses and the values of the corner frequencies are shown for the I/D and TBS models. *Right column*: phase responses of the transfer functions. The I/D and PhaseSync models both have a  $\pi/2$  phase response. Initial

5 Right column: phase responses of the transfer functions. The I/D and PhaseSync models both have a  $\pi/2$  phase response. Initial Butterworth filters has removed frequencies lower than 1/8000 yr<sup>-1</sup>, the effect of which is represented as the gap at the low frequencies of the amplitude and phase responses.

Figure 3 a. Simulation of the Antarctic EDC record based on the NGRIP record from Greenland. Both EDC and NGRIP have been band-pass filtered, normalized, and then tapered. The input NGRIP record is then convolved with each model transfer function. The results are labeled as their model names "I/D", "TBS", and "PhaseSync". The τ value was set to 1120 yrs for the 5 TBS model as in (Stocker and Johnsen 2003). EDC ice core record (in red) is shown at the bottom as a reference for visual comparison. b. Simulation of the Greenland NGRIP record based on the EDC record from the Antarctic. Both EDC and NGRIP have been processed in the same way as in (a). Then the input EDC record is used to convolve with transfer functions from the inverse of the polar teleconnection models that are labeled as "I/D", "TBS", and "PhaseSync". The NGRIP (in red) is shown at the bottom as a reference. c-d. cross-correlation functions between each of the model results and the target record (EDC in c and NGRIP in d).

Figure 4 a. Model skills in reproducing the Antarctic record measured in the Pearson correlation coefficients between the simulations and the actual EDC record. The skill of the TBS model depends on the value of  $\tau$ . b-c. Model skills represented by correlation coefficients between the simulations and the actual NGRIP record. b. The dependence of model skill on the corner frequency  $\sigma$  of the low-pass filter while fixing  $\tau$  at four different values ranging from 500 to 6000 yrs. c. The dependence of model skill on the time constant  $\tau$  from the TBS model while fixing  $\sigma$  at 1/1000 yr<sup>-1</sup>.

Figure 5 Model skills when taken into account both directions.

Figure A1 Illustration of the phase shift concept. s1 and c1 are sine and cosine function both with frequency 0.5, but with a phase difference of  $\pi/2$  between them; the same is true with s2 and c2, which have double the frequency (f = 1) compared to the s1 and c1 pair. While a simple time shift of one quarter of their corresponding period can transform s1 to c1 or s2 to c2, their respective sums (s1 + s2) and (c1 + c2), still have a phase shift of  $\pi/2$ , cannot be transform into each other by simple shift in time. As in the case of the polar climate records, the sums from sine and cosine function have very distinct shape characteristics.

Figure A2 Frequency response comparison between the I/D and TBS models.

Figure A3 Simulation of Greenland NGRIP record from the Antarctic EDC record when the input signal did not go through the low-pass filtering.

# Tables

| Direction         | Model type  | HP filter | LP filter | Reference            | Remarks                 |
|-------------------|-------------|-----------|-----------|----------------------|-------------------------|
|                   | I/D         | 10 ky     | NA        | Roe & Steig, 2004    |                         |
|                   | TBS         | 8 ky      | NA        | Stocker &            | No mention of LP filter |
| $G \rightarrow A$ |             |           |           | Johnsen, 2003        |                         |
|                   | PhaseSync + | ~10 ky    | ~10 ky    | Oh et al., 2014      | EMD, SSA, and linear    |
|                   | VDP         |           |           |                      | filter used             |
|                   | I/D         | MW: 7     | MW: 700   | Barker et al., 2011  | MW: moving average      |
| $A \rightarrow G$ |             | ky        | yr        |                      | window width            |
|                   | TBS         | 8 ky      | 1 ky      | Siddall et al., 2006 | Gaussian filter used    |

Table 1. Summary of types of filters and their parameters used in data pre-treatment for the conceptual models discussed in this study. EMD: empirical mode decomposition; SSA: singular spectrum analysis; MW: moving window average; LP: lowpass filter; VDP: Van day Bal agaillator  $C_{\rm exp}$  As reconstruction of Antomatic record based on the tof the Greenland  $A_{\rm exp}$  of the records

5 VDP: Van der Pol oscillator.  $G \rightarrow A$ : reconstruction of Antarctic record based on that of the Greenland;  $A \rightarrow G$ : the reverse.

| Model     | Time domain                                                                                | Frequency domain                     | Description |
|-----------|--------------------------------------------------------------------------------------------|--------------------------------------|-------------|
| I/D       | $-\frac{1}{\Delta t}$                                                                      | $\frac{1}{j2\pi k\Delta f}$          |             |
| TBS       | $-e^{-\frac{i\Delta t}{\tau}}$                                                             | $\frac{\tau}{1+j2\pi k\Delta f\tau}$ |             |
| PhaseSync | $\left(\frac{2}{N}\right)\sin\left(\frac{i\pi}{2}\right)^2\cot\left(\frac{i\pi}{N}\right)$ | $-j sgn\left(\frac{N}{2}\right)$     | N is even   |
|           | $(1/N)[\cot(i\pi/N)$                                                                       | (k) - k sgn(k)                       | N is odd    |
|           | $-\cos(i\pi)/\sin(\frac{i}{N}\pi)$ ]                                                       |                                      |             |

Table 2. Transfer functions for the conceptual models. For the time domain representation, i is the index number ranging from 1 to the length of the record N. For the frequency domain representation, j is the square root of -1 and k is the index of the frequency. See Figure 2 for visualization of the transfer functions, their amplitude and phase responses.

5