# Peer review of "Teleconnection between the climates of the Polar Regions during the last ice age"

_Climate of the Past, 2018_

## Referee Comment (RC1) · Anonymous Referee #1 · 14 Jun 2018

Yang and Rial compare three conceptual mathematical models of the bipolar coupling of millennial-scale climate variability, i.e. the relationship between Greenland Dansgaard-Oeschger (D-O) and Antarctic Isotopic Maxima (AIM) events. They use published ice core water stable isotope data (NGRIP and EDC) on previously established chronologies. They compare the transfer functions for each of these models, and evaluate how well the models can predict the Antarctic record given the Greenland record (G-to-A), and vice versa, how well the models can predict the Greenland record given the Antarctic record (A-to-G). They find that all three models are essentially equivalent for the G-to-A case, and that the PhaseSync method works best for the A-to-G case.

Based on this the authors argue that the PhaseSync method is the best framework to

describe the bipolar coupling, a conclusion with which I disagree (for reasons outlined below). Overall, the study does not contribute much to the literature, given that it is basically a (not very comprehensive) review of published models and no new dynamical insights are presented. One gets the feeling that the authors were aiming at the so-called "least publishable unit". I make a suggestion below on a natural way in which the study could be extended that provides more value to the literature.

My main criticism is that the conclusion that the PhaseSync method is superior over the I/D and TBS models is not valid.

First, this conclusion is based purely on the A-to-G test; in the G-to-A test, the I/D and TBS models actually score better. A recent paper by the WAIS Divide Project Members (2015), has clearly demonstrated the N-to-S directionality of the teleconnection. Coupled atmosphere-ocean GCMs clearly reproduce the SH signals after applying NH perturbations to the AMOC (Pedro et al., 2016). Since there is data- and model-based evidence that the teleconnection has a N-to-S directionality, it is obvious that G-to-A (and not A-to-G) is the most relevant test for these models.

The I/D and TBS models were designed around the assumption of a N-to-S teleconnection, and therefore Yang and Rial test these models outside of the scope intended by their authors.

Second, the PhaseSync model may perform well at the A-to-G test, to my understanding it is just a mathematical construct without a firm basis in climate dynamics. In the PhaseSync model, each polar region represents an independent climate oscillator, which are then synchronized like pendulum clocks. This implies that the oscillations would be present also in the absence of the teleconnection – simply not in a synchronized way. It seems unlikely to me that Greenland temperature could fluctuate by > 10 degree C in the absence of the coupled sea ice and AMOC perturbations (i.e. the teleconnection mechanism). I have not seen any realistic description of what these uncoupled oscillators are. By contrast, there is a wide range of paleoceanographic evidence linking the D-O cycle to variations in Atlantic circulation (Lynch-Stieglitz, 2017), which is the physical basis/justification of the I/D and TBD models (which are identical for most practical purposes, as the authors also suggest). The I/D and TBD concepts are further well simulated in climate models (Ganopolski and Rahmstorf, 2001; Pedro et al., 2016; Pedro et al., 2018; Schmittner et al., 2003). This wide gap in supporting evidence should be acknowledged, and weighed in the evaluation of what model is most realistic.

As I mentioned, the paper in its current form does not have much new to offer to the paleoclimate community. The observation that the PhaseSync model is superior in the A-to-G reconstruction does open the opportunity to redo the analysis of (Barker et al., 2011), and provide an improved 800,000 year reconstruction of Greenland climate. Such a reconstruction would be a valuable product, that naturally extends from the presented work. The long-term orbital component would have to be added to the millennial component, but that should be straightforward.

Throughout the manuscript would benefit from editing for grammar and sentence structure.

Detailed comments:

P1L23: Note that Broecker 1998 is talking about a different type of seesaw, one in deepwater formation.

P2L2: Note that the I/D model also suggests a pi/2 phase shift, and so this observation should not be solely accredited to Oh et al (2014), as many other studies have precedence.

P2L5-10 and elsewhere: Please elaborate on what the individual oscillators would be without the teleconnection. And what physical mechanisms are responsible for the coupling? Simply stating "ocean and atmospheric teleconnections" is too vague.

P3L4: these are not the correct references for this statement. See some of the references given above are more suitable.

P4L25: note the similarity between the I/D and TBD models, and the fact that they are well supported by sediment records and climate models.

P6 section 2.3. There is a big misunderstanding here. The authors argue that previous studies were hampered by using N-S synchronization, whereas this uncertainty is eliminated by using the AICC2012 chronology for both cores. However, the AICC2012 chronology is constructed via N-S synchronization! So the same uncertainties apply. Please rewrite this section. Also, the WAIS Divide ice core is much more accurately synchronized to Greenland, and would provide the better data set to test these ideas on.

P8L7: Note that the TBD model also gives a pi/2 phase shift at the relevant frequencies (∼1ka).

P8 section 3.3: note that the A-to-G test is not relevant for testing the models, as the data are clear that the teleconnection operates N-to-S

P9L9: Buizert et al. (2015) should be cited as WAIS Divide Project Members (2015)

P11L3-9: another, more traditional explanation for the great Antarctic warmth during Heinrich stadials, is that these stadials are lengthened by iceberg delivered freshwater, allowing more heat to build up in the SH via the BPS.

P11L19-25: Note that there is a lot of ocean sediment and climate model evidence for the TBS and AMOC variations on which it rests, such evidence is currently lacking for the PhaseSync.

P128: "suggest potential physical coupling mechanisms". Which ones? These are missing from the theory, as far as I can tell.
* * *

---

## Referee Comment (RC2) · Anonymous Referee #2 · 20 Jul 2018

Review of "Teleconnection between the climates of the Polar Regions during the last ice age" by Yang and Rial. This paper compares three different conceptual models that describe the connection between Greenland and Antarctic isotopic temperature proxies during the last glacial period. The authors favor their "PhaseSync" model, in which the two poles are phase synchronized, over two alternatives: the favored thermal bipolar seesaw model, and an integration/differentiation model. The paper is well written and includes a good amount of detail. The paper is interesting because it offers a contrarian model to the preferred point of view. In principle at least, their model can be tested against paleoclimate evidence and evaluated.

The three models are compared here using a Pearson product-moment correlation coefficient between the derived and real opposite-pole records. Basically, the wiggles

derived using one model are somewhat better than the wiggles derived using the other models. The paper also discusses other weaknesses of the alternatives, such as parameter sensitivity. By itself, the modest correlation improvement that the authors find in this paper (∼0.7 versus ∼0.6) could have a number of explanations that are not pertinent evidence proving superiority of their model. So, by itself, the paper does not present sufficient evidence, no "smoking gun", to justify adopting this model and rejecting the others. The paper does reference arguments in previous papers, e.g. Rial (2012) and Yang et al. (2014), and perhaps these papers are meant to hang together.

The authors mention ice core chronology, but do not delve into the details of how these chronologies are determined, which would seem relevant here. Examination of individual climate events, as opposed to secular records, might provide better discrimination between models. Perhaps there are specific climate change instances where the PhaseSync model clearly outperforms. Perhaps close tracking of the energy in the ocean/atmosphere, using physical evidence, could provide a smoking gun. Without clear evidence, the authors will have an uphill battle overcoming the intransigence of scientists.

Typos:

Pg. 2, line 7: "Such [a] polar synchronization hypothesis..."

Pg. 4, lines 6-7: "...have also been used to [obtain] the first order approximation of Greenland climate history beyond the [extent] of its ice core record."

Pg. 5, line 18: "different studies have different definition of millennial scale variation in terms of what frequency band was to include in the filtered data that would be feed into the models."

Pg. 6, line 24: "...which means that one can implement the model via using polar climate records.."

Pg. 7, line 20: "...when the tau is in the range of hundreds of years..."

Pg. 8, line 10: "...the implementation of each model was carried out by convolving [each] polar record with each transfer function."

Pg. 11, line 2: "the HT polar synchronization model performs much better than [either] of its main competitors, [the] I/D and TBS models.

---

## Author Comment (AC1) · 23 Aug 2018

We would like to start our response by thanking both reviewers for their efforts in evaluating and commenting on our manuscript. The reviewers have offered constructive suggestions for a comprehensive and objective discussion of the models to not only base on their skills to reproduce climate records, as done in this study, but also on the abundance and strength of the supporting evidence from past literature. We will integrate these changes into the revised manuscript. At the same time, we found and would like to further clarify some of the misunderstandings in the reviewers' comments, especially the ones associated with the physical mechanisms of the PhaseSync model and with the concept of phase synchronization itself. We hope that through the following clarifications, we have resolved some of the major criticisms from the reviewers.

Response to Reviewer 1:

The criticism from "Anonymous Referee #1", or AR1, focused on three points. First AR1 argued that the content of this research is of little contribution to the literature, stating the authors aimed at a "minimum publishable unit". Then, AR1 argued that the I/D and TBS models are based on N-S directionality of the polar teleconnection, thus applying or testing them in the S-N, that is to reconstruct the Greenland records from the Antarctic ones, falls outside the scope of the models. Third, AR1 argued that the different strength in paleoclimate evidence supporting the models should be acknowledged along with their ability to reconstruct the existing records. In the following text, we will address the three criticisms from AR1.

First, it may appear that a model comparison study to conclude which model is superior among three which produce close results does not advance science much, hence the criticism "minimum publishable unit". The problem here is that some of the models we contrast against each other have already entered the scientific literature as the established, if not the only, models defining climate dynamic between the poles (for example, the TBS model was included in a textbook on climate modeling (Stocker, 2011)). Further, Markle et al. (2016) in an article about teleconnections during D-O events, treated the I/D model as if it was the obvious and correct one, ignoring all other models.

This is why we could not agree with the characterization of our paper as a "minimum publishable unit". Our paper is self-contained, its objectives are clearly independent of our published record, and its results important to be known, especially since it shows an alternative mechanism, as also being pointed out by referee 2. AR1 writes that our paper offers "no new dynamical insights". We would respectively argue that, quite the contrary, phase synchronization is a new dynamical explanation of the polar climate fluctuations that is consistent with the data and modeling results (Rial, 2012; Rial and Saha, 2011).

Second, while initially the TBS models and I/D models were proposed or tested under Greenland to Antarctica direction (or N-S direction), such construct of the models does not prohibit reconstructing records in the reverse direction, provided that the defining models demonstrate a one to one relationship between polar climates, a condition which all three models satisfy. In fact, TBS and I/D models have been used to reconstruct or extend Greenland records based on the much longer Antarctic one (Barker et al., 2011; Siddall et al., 2006). It seems to us that by invalidate testing the TBS and I/D models in the S-N direction, as suggested by AR1, automatically invalidates their usage to reconstruct the Greenland climate. Therefore, the PhaseSync model naturally becomes the only one of the three models that has been constructed with the interactive role of polar teleconnection in mind, thus should be valid for reconstructing the Greenland record as it assumes bidirectional coupling.

AR1 has also argued that the PhaseSync (Rial 2012) model was not built upon physical processes. We respectfully disagree with AR1 and present the physics of the PhaseSync model below. The PhaseSync model was originally proposed to describe and model the polar climate interaction for the abrupt millennial scale events during the last glacial period (Rial, 2012). It was built as an extension of a van der Pol oscillator (Saltzman, 2002) that has strong physical support and was originally constructed by Saltzman et al. (1981). This model closely reproduced the entire GRIP record and closely simulated the sea ice extent (including large relative amplitudes) and average oceanic temperature obtained with the much larger and detailed ECBilt-Clio (Rial and Saha, 2011), a GCM of intermediate complexity (Goosse and Fichefet, 1999). Building upon this, the PhaseSync model was constructed by using two van der Pol oscillators (one for each polar region), coupled through the temperature difference and heat storage of the ocean (Rial, 2012). For details of this model and its applications, interested readers should refer to the following literature(Oh et al., 2014; Rial, 2012; Rial and Saha, 2011; Yang et al., 2014).

We also disagree with AR1's description of oscillator behavior in the absence of the

phase synchronization. AR1's comments suggest that the oscillatory amplitude would stay the same before the polar climates were synchronized. This is not true as phase synchronization does not necessarily corresponds to changes in amplitude. Such phenomenon has been documented widely (Balanov et al., 2008; Maraun, 2005; Pikovsky et al., 2003) and in fact, these authors described it as follows: phase synchronization results when two or more nonlinear oscillators couple and therein adjust their (initially different) natural rhythms to a common frequency and constant relative phase, while amplitudes are not necessarily correlated. So having strong D/O events during the synchronized state of polar climates does not imply amplitude of the same strength for the individual oscillators when the synchronization is lacking.

Third, AR1 suggested that, a comprehensive intercomparison study needs to consider the supportive literature behind each model. We appreciate the reviewer's suggestion such information will be added when revising the manuscript. We also appreciate the reviewer's suggestion of reconstructing a 800,000 year Greenland climate using the PhaseSync model. However, it feels a bit out of place to us to include a climate reconstruction in a model intercomparison paper.

To conclude, we appreciate the constructive suggestions AR1 has provided, pointing us to the gap of our literature review for some of the models. However, we disagree with most of the reviewer's criticisms. To our knowledge, our study is the first one to both theoretically and numerically compares three prominent conceptual models that describe the links between the abrupt climate changes registered in records from both polar regions. As all three models have their own extensive supporting studies, we tried to be objective in our intercomparison methods in this study, so that the results can serve as a starting point for future refinement of these models or, as foundation upon which new conceptual models can be established.

References

Balanov, A., Janson, N., Postnov, D. and Sosnovtseva, O.: Synchronization: From

Simple to Complex, Springer Science & Business Media. [online] Available from: https://market.android.com/details?id=book-Fx_6pPCxXWAC, 2008.

Barker, S., Knorr, G., Edwards, R. L., Parrenin, F., Putnam, A. E., Skinner, L. C., Wolff, E. and Ziegler, M.: 800,000 years of abrupt climate variability, Science, 334(6054), 347–351, doi:10.1126/science.1203580, 2011.

Goosse, H. and Fichefet, T.: Importance of ice-ocean interactions for the global ocean circulation: A model study, J. Geophys. Res., 104(C10), 23337–23355, doi:10.1029/1999JC900215, 1999.

Maraun, D.: Epochs of phase coherence between El Niño/Southern Oscillation and Indian monsoon, Geophys. Res. Lett., 32(15), 274, doi:10.1029/2005GL023225, 2005.

Markle, B. R., Steig, E. J., Buizert, C., Schoenemann, S. W., Bitz, C. M., Fudge, T. J., Pedro, J. B., Ding, Q., Jones, T. R., White, J. W. C. and Sowers, T.: Global atmospheric teleconnections during Dansgaard–Oeschger events, Nat. Geosci., 10(1), 36–40, doi:10.1038/ngeo2848, 2016.

Oh, J., Reischmann, E. and Rial, J. A.: Polar synchronization and the synchronized climatic history of Greenland and Antarctica, Quat. Sci. Rev., 83, 129–142, doi:10.1016/j.quascirev.2013.10.025, 2014.

Pikovsky, A., Rosenblum, M., Kurths, J. and Kurths, J.: Synchronization: A Universal Concept in Nonlinear Sciences, Cambridge University Press. [online] Available from: https://market.android.com/details?id=book-FuIv845q3QUC, 2003.

Rial, J. A.: Synchronization of polar climate variability over the last ice age: in search of simple rules at the heart of climate's complexity, Am. J. Sci., 312(4), 417–448, doi:10.2475/04.2012.02, 2012.

Rial, J. A. and Saha, R.: Modeling Abrupt Climate Change as the Interaction Between Sea Ice Extent and Mean Ocean Temperature Under Orbital Insolation Forcing, in Geophysical Monograph Series, pp. 57–74., 2011.

Saltzman, B.: Dynamical paleoclimatology, volume 80 of International Geophysics Series, 2002.

Saltzman, B., Sutera, A. and Evenson, A.: Structural Stochastic Stability of a Simple Auto-Oscillatory Climatic Feedback System, J. Atmos. Sci., 38(3), 494–503, doi:2.0.CO;2">10.1175/1520-0469(1981)038<0494:SSSOAS>2.0.CO;2, 1981.

Siddall, M., Stocker, T. F., Blunier, T., Spahni, R., McManus, J. F. and Bard, E.: Using a maximum simplicity paleoclimate model to simulate millennial variability during the last four glacial periods, Quat. Sci. Rev., 25(23-24), 3185–3197, doi:10.1016/j.quascirev.2005.12.014, 2006.

Stocker, T.: Introduction to Climate Modelling, Springer Science & Business Media. [online] Available from: https://market.android.com/details?id=book-D4zulgFb5JwC, 2011.

Yang, X., Rial, J. A. and Reischmann, E. P.: On the bipolar origin of Heinrich events, Geophys. Res. Lett., 41(24), 9080–9086, doi:10.1002/2014gl062078, 2014.

---

## Author Comment (AC2) · 23 Aug 2018

We would like to start our response by thanking both reviewers for their efforts in evaluating and commenting on our manuscript. The reviewers have offered constructive suggestions for a comprehensive and objective discussion of the models to not only base on their skills to reproduce climate records, as done in this study, but also on the abundance and strength of the supporting evidence from past literature. We will integrate these changes into the revised manuscript. At the same time, we found and would like to further clarify some of the misunderstandings in the reviewers' comments, especially the ones associated with the physical mechanisms of the PhaseSync model and with the concept of phase synchronization itself. We hope that through the following clarifications, we have resolved some of the major criticisms from the reviewers.

Response to Reviewer 2:

"Anonymous Referee #2", or AR2, has mainly argued that by comparing the correlation between the model simulations and the actual records provides not sufficient evidence to adopt the PhaseSync model over the others.

AR2 was correct in observing the small increase from ∼0.6 to ∼0.7 in correlation when reproduce a particular frequency band of the Greenland record using the PhaseSync model (Figure 3d). However, the models' ability to reproduce records are fully explored in the next figure (Figure 4b) where various possible parameter values are tested. Depending on the parameter values, the increase of correlation from using PhaseSync model can be as big as 0.4 (Figure 4). Such difference is further summarized in Figure 5, where model skills in reproducing polar records from both poles are compared.

As AR2 noticed, the PhaseSync model, described in this study, has a natural lineage from previous publications (Oh et al., 2014; Rial, 2012; Yang et al., 2014). And to fully understand the model, these literature needs to be reviewed together. We would be happy to include key points for this model in the revised version of the manuscript to make this study better self-contained.

AR2 also suggested the authors to include more details on the chronology used. It is true that the chronology is critical to our analysis. We have used only published AICC2012 chronology (Veres et al., 2013) and given citation wherever appropriate. However, as a clarification, we will add detailed description of the chronologies used in this study in the revised version of the manuscript.

AR2 also kindly suggested us to closely compare the models for certain individual events to gain insight on the relative skills of the models. The numeric tests in this study were designed to assess the models' ability to reproduce millennial scale variabilities, which all three models were originally proposed for explaining. On the contrary, comparing model skills for certain millennial-scale events likely involves components of the records that belong to centennial or even higher frequency bands, which the models

may be unfit for.

References

Oh, J., Reischmann, E. and Rial, J. A.: Polar synchronization and the synchronized climatic history of Greenland and Antarctica, Quat. Sci. Rev., 83, 129–142, doi:10.1016/j.quascirev.2013.10.025, 2014.

Rial, J. A.: Synchronization of polar climate variability over the last ice age: in search of simple rules at the heart of climate's complexity, Am. J. Sci., 312(4), 417–448, doi:10.2475/04.2012.02, 2012.

Veres, D., Bazin, L., Landais, A., Toyé Mahamadou Kele, H., Lemieux-Dudon, B., Parrenin, F., Martinerie, P., Blayo, E., Blunier, T., Capron, E., Chappellaz, J., Rasmussen, S. O., Severi, M., Svensson, A., Vinther, B. and Wolff, E. W.: The Antarctic ice core chronology (AICC2012): an optimized multi-parameter and multi-site dating approach for the last 120 thousand years, Clim. Past, 9(4), 1733–1748, doi:">10.5194/cp-9-1733-2013>, 2013.

Yang, X., Rial, J. A. and Reischmann, E. P.: On the bipolar origin of Heinrich events, Geophys. Res. Lett., 41(24), 9080–9086, doi:10.1002/2014gl062078, 2014.